# Thyroid Axis and Vestibular Physiopathology: From Animal Model to Pathology

**DOI:** 10.3390/ijms24129826

**Published:** 2023-06-06

**Authors:** Guillaume Rastoldo, Brahim Tighilet

**Affiliations:** 1Aix Marseille Université-CNRS, Laboratoire de Neurosciences Cognitives, LNC UMR 7291, 13331 Marseille, France; guillaume.rastoldo@inserm.fr; 2GDR Vertige CNRS Unité GDR2074, 13331 Marseille, France

**Keywords:** thyroxine, thyroid hormones, thyroid axis, vestibular system, vestibular compensation, vertigo

## Abstract

A recent work of our group has shown the significant effects of thyroxine treatment on the restoration of postural balance function in a rodent model of acute peripheral vestibulopathy. Based on these findings, we attempt to shed light in this review on the interaction between the hypothalamic–pituitary–thyroid axis and the vestibular system in normal and pathological situations. Pubmed database and relevant websites were searched from inception through to 4 February 2023. All studies relevant to each subsection of this review have been included. After describing the role of thyroid hormones in the development of the inner ear, we investigated the possible link between the thyroid axis and the vestibular system in normal and pathological conditions. The mechanisms and cellular sites of action of thyroid hormones on animal models of vestibulopathy are postulated and therapeutic options are proposed. In view of their pleiotropic action, thyroid hormones represent a target of choice to promote vestibular compensation at different levels. However, very few studies have investigated the relationship between thyroid hormones and the vestibular system. It seems then important to more extensively investigate the link between the endocrine system and the vestibule in order to better understand the vestibular physiopathology and to find new therapeutic leads.

## 1. Introduction

The regulation of thyroid hormone production is under the control of the hypothalamic–pituitary–thyroid (HPT) axis (Figure 1). TRH (Thyrotropin-Releasing Hormone), which is synthesized and secreted by the neurons of the paraventricular nuclei of the hypothalamus, stimulates the release of TSH (Thyroid-Stimulating Hormone) by the pituitary gland. TSH binds to its membrane receptor in the thyroid follicular cells and triggers the synthesis and secretion of the following thyroid hormones: Thyroxine (Tetraiodothyronine, T4) and T3 (Triiodothyronine). When the concentration of T4 and T3 in the blood increases, a negative feedback loop is set up to inhibit the pituitary response to TRH and decrease TSH secretion.

Thyroid hormones are released into the blood largely as T4 and are then transported into the plasma by thyroxine-binding globulin, albumin or transthyretin [1]. Active transport of thyroid hormones in the rat brain is mediated primarily by MCT8 (Monocarboxylate Transporter-8) and OATP1 (Organic Anion Transporter Protein-1), as well as transthyretin in the choroid plexuses [2,3,4]. The major biologically active form of thyroid hormones is T3, which is produced from the pro-hormone T4 by the enzyme D2 (Iodothyronine Deiodase type 2). This enzyme is mainly expressed in astrocytes and tanycytes lining the third ventricle [5]. Moreover, particularly high levels of D2 mRNA have been detected in the vestibular nuclei of guinea pig fetuses [6]. Thus, based mainly on immunohistochemical studies, the mode of action of thyroid hormones in the brain would be the following (Figure 2):

First, T4 (and to a lesser extent T3) is transported across the blood–brain barrier by OATP1 or MCT8. T4 is then transported into astrocytes by a still unknown transporter. In astrocytes, T4 is converted to T3 by D2. Biologically active T3 is then transported out of astrocytes by another unknown transporter. Neuronal uptake of T3 is facilitated by MCT8 and then T3 exerts its genomic action by binding to its receptor. Finally, T3 is inactivated by D3 (Iodothyronine Deiodase type 3) to T2 (diiodothyronine). D3 also catalyzes the deiodination of T4 to rT3 (reverse T3), an inactive form of T3 [3,7,8,9,10].

The action of thyroid hormones at the genomic level is mediated by nuclear thyroid hormone receptors (TRs) that activate or repress the expression of target genes. Some genes that could be identified include cell cycle regulatory genes Sox2, cyclin D1 and c-myc [11,12] and genes involved in neural progenitor development, such as Tis21, Tlx, Dlx2, Math-1 and Ngn1 [13]. 

The TR alpha and beta genes generate several distinct TR isoforms, including TRα1 (dominant isoform in the brain with 70% expression), TRα2, TRβ1 and TRβ2 expressed in the mammalian brain. Both TRα1 and TRα2 are widely expressed in the adult brain [14,15] with high levels in vestibular nuclei found in guinea pig fetuses [6]. The TRβ isoforms on the other hand show increasing levels of expression throughout postnatal development. TRβ1 is ubiquitously expressed in the brain, while TRβ2 expression is restricted to the hypothalamus and the hypophysis. However, this mode of action is incomplete and many studies support the existence of additional mechanisms of action [16,17,18,19]. In particular, studies have shown that thyroid hormones can sometimes act within minutes, a time frame of action that is far too rapid for a gene transcription-mediated response. Indeed, thyroid hormones can have immediate actions by binding to extranuclear receptors, including TRα and TRβ, located in the cytoplasm, cell membrane integrin αVβ3 receptor, cytoskeleton, and mitochondria, modulating many intracellular pathways [17,18,19,20]. In addition, T4 can have a more potent effect than T3 [21]. These alternative pathways are generally referred to as “non-genomic pathways” but a better nomenclature of the TH actions might be necessary to better classify them [21]. Genomic and non-genomic actions of thyroid hormone are represented in Figure 3.

## 2. Thyroid Hormones in the Development of the Inner Ear

Interestingly, congenital hypothyroidism (CH) results in delayed inner ear development. Many studies have focused on alterations in auditory function [22,23,24] and in comparison, alterations in vestibular function in congenital hypothyroidism remain under-explored [25,26]. Regarding auditory function, CH delays the development of the organ of Corti and the functional maturation of the cochlea. The outer hair cells are poorly differentiated and the tectorial membrane is thinned, resulting in decreased hearing and in the most severe forms of congenital deafness. Meza and collaborators studied more precisely the vestibular functions in newborn rats with congenital hypothyroidism. They were able to demonstrate that the post-rotatory nystagmus normally evoked after animal rotations was absent in CH animals. This result confirms the previous study of the same team showing the absence of a response after stimulation of the semicircular canals in CH rats [27]. Daily intramuscular L-T4 injections into the quadriceps from P12 in CH rats restored the responses of the semicircular canals at P27 and the presence of a post-rotatory nystagmus identical to control animals [26]. At the cellular level, CH induces a delay in the differentiation of vestibular hair cells, the formation of calyx synapses and the myelination of primary vestibular axons [25]. These findings highlight the importance of thyroid hormones in the developmental and maturation processes of the inner ear, which are necessary for adequate functioning of the adult vestibular system.

## 3. A Vestibular Modulation of the Thyroid Axis?

In a quite interesting way, gravity influences thyroid cells (for review: [28]). Indeed, thyroid carcinoma cells cultured at 0 g to simulate microgravity strongly reduce the secretion of free T4 and T3 [29]. These data are in agreement with the decrease in T4 and T3 levels of astronauts after a space flight [30,31,32,33]. Morphological examination of the thyroid glands of rats exposed to a space flight has shown histological changes indicating a reduction in thyroid activity [34,35]. Conversely, in rats subjected to hypergravity, T3 production is increased [36]. These different effects have not been characterized precisely and further studies are needed to clarify the role of gravity on thyroid function. It is possible that the vestibular system, activated by terrestrial gravity, has connections to the thyroid gland. 

We can make an interesting analogy between the data concerning the impact of microgravity on thyroid function and the effects of space flight on the muscles involved in posture. Since the mid-1970s, it has been recognized that spaceflight induces substantial muscle atrophy, particularly in the antigravity and postural muscles [37]. Indeed, previous studies have shown in rats a loss of muscle mass of the soleus muscle of about 30% in microgravity [38,39,40,41]. In addition, microgravity exposure produces structural changes in soleus muscle with a shift to a faster phenotype, correlated with a significant decrease in type 1 and 2A fibers and an increase in 2X and 2B fibers, as well as changes in myosin heavy chain isoforms [42]. Given the influence of gravity on the thyroid system, it cannot be ruled out that the alteration of the soleus muscle (related to its underutilization) in a hypogravity environment also involves this hormonal component [43]. Indeed, the crew of the International Space Station devotes an average of 2.5 h per day to physical exercise; however, even this is not sufficient to compensate for the effects of continuous exposure to microgravity on the muscular system [44]. These results suggest the involvement of additional mechanisms in the alteration of antigravity muscles in microgravity. Considering the presence of TH receptors in skeletal muscles and the involvement of TH in the regulation of muscle tone [45], it is very likely that the histological alterations of the soleus muscle observed in astronauts or rats in microgravity result from the hypofunction of the thyroid gland in microgravity. This point is developed further in Section 5.7 below.

At the central level, we can also assume connections between the vestibular nuclei and the neurons of the hypothalamus responsible for TRH release. Indeed, the neurons that synthesize and release TRH are located in the paraventricular nucleus (PVN) of the hypothalamus [46]. Electrical and caloric stimulation of vestibular pathways elicits a response in PVN neurons in guinea pigs [47,48]. Retrograde tracing has also demonstrated the presence of a direct vestibulo-paraventricular projection in rats [49] and a paraventricular-vestibular pathway has also been described [50]. These neuroanatomical pathways support the link between the vestibular system and the stress axis or hypothalamic–pituitary–adrenal axis [51,52]. The interaction between the vestibular system and thyroid axis remains unexplored. The vestibular syndrome after unilateral vestibular deafferentation activates the stress axis [53,54]. It would be interesting to demonstrate that a change in vestibular system activity generated either by the stimulation of vestibular receptors or by their suppression activates the thyroid axis as is the case for the stress axis and the neuronal histaminergic axis [55].

## 4. Thyroid Disorder and Dizziness in Humans: Is There a Link?

Despite the histopathological parameter correlated with Meniere’s disease (endolymphatic hydrops: dilation of the membranous labyrinth of the inner ear) [56,57], its etiopathogenesis remains uncertain and multifactorial. Autoimmune factors, trauma, viral infection, genetic predisposition, hormonal disorder, and metabolic factors could contribute to the development of Meniere’s disease [58]. The possible correlation between hypothyroidism and Meniere’s disease was proposed over 40 years ago. Powers and colleagues reported hypothyroidism in 17% of 98 patients with Meniere’s disease [59]. A recent study of 5410 hypothyroid patients demonstrates that subjects with hypothyroidism have a greater risk of developing Meniere’s disease than euthyroid subjects [60]. Bhatia and colleagues report that symptoms of vertigo were observed in 29.1% of patients with hypothyroidism [61]. More recently, Kim et al. demonstrated that both hypo- and hyperthyroidism were related to Meniere’s disease [62]. The attenuation of Meniere’s disease after thyroxine supplementation of hypothyroid patients is controversial. In Powers’ study, only 3 of 97 patients had an improvement in symptoms after thyroxine treatment [59]. In another study, 12 of 35 hypothyroid patients were found to have Meniere’s disease and all 12 patients reported subjective improvement in symptoms after 12 weeks of thyroxine treatment [63]. According to Lin’s population-based study, the overall incidence of Meniere’s disease was lower in hypothyroid patients who received thyroxine treatment compared to those who did not; however, the difference was not significant [60].

Other studies have focused more specifically on the links between vertigo and autoimmune hypothyroidism, such as Hashimoto’s thyroiditis (for review: [64]). Indeed, inflammatory diseases (such as Hashimoto’s: chronic autoimmune inflammation of the thyroid causing hypothyroidism) could cause a cross-immune reaction against inner ear cells and impair cochlear and vestibular functions. Following this reasoning and considering the inflammatory basis of Hashimoto’s thyroiditis, it is possible to find a relationship between the two diseases. In support of this hypothesis, Kim et al. studied the composition of the endolymphatic sac in a group of 13 patients with Meniere’s disease and found the presence of immunoglobulins, proving the possibility of immune reactions in the labyrinth [65]. Fattori and colleagues report that the prevalence of anti-thyroid autoantibodies was significantly higher in the group of Meniere’s disease patients (38%) than in the two control groups (7% in a healthy control group, and 12% in a group of non-Meniere’s disease patients). These data on 50 Meniere’s patients indicate a close relationship between autoimmune thyroid disease and Meniere’s disease [66]. This study reinforces the hypothesis of a possible pathogenic role of autoimmunity in the development of Meniere’s disease [66]. These data have been confirmed by other teams and confirm that patients with Meniere’s disease or benign paroxysmal positional vertigo (BPPV) are potential candidates to develop Hashimoto’s thyroiditis and vice versa [67,68,69]. 

## 5. Thyroid Hormones and Vestibular Compensation in Animal Models of Vestibulopathy

Unilateral vestibular loss causes a vestibular syndrome in most species, including humans. This syndrome is characterized by a cascade of functional disorders, including postural imbalance, altered gait, spontaneous nystagmus, altered vestibulo-ocular reflexes, and cognitive and neurovegetative disorders (Figure 4). The neurophysiological mechanisms at the origin of the vestibular syndrome were identified from the 1960s onwards, on the basis of electrophysiological recordings made at the level of the vestibular nuclei (VN) of the brainstem of vestibulo-injured animals [70,71]. These studies demonstrated that the neurophysiological support of the vestibular syndrome resulted from a disruption of the electrophysiological balance between the ipsi- and contralateral VN. This asymmetry of electrical activity between the VN disrupts the vestibulospinal and vestibulo-ocular reflexes that cause postural and oculomotor deficits, and also alters the vestibulocortical signals that cause perceptual and cognitive deficits [72,73].

The different symptoms that constitute the vestibular syndrome decline progressively, each with its own kinetics, generally leading to a rapid and complete disappearance of static deficits and an often slower and incomplete regression of dynamic deficits. This phenomenon is referred to as vestibular compensation [73,74,75]. Closely related to the restoration of vestibular function are two major and interrelated neurobiological phenomena that occur in the deafferented VN of the brainstem. The first is the spontaneous recovery of the electrophysiological balance between homologous VN [75,76,77,78,79]. Our group was able to demonstrate that the expression of excitability markers such as the cotransporter KCC2 [80,81], or even the SK channels [82], are modulated in this area, throughout the induction and compensation phases of the vestibular syndrome. In both cases, pharmacological actions targeting these excitability markers have been shown to reduce the intensity of the vestibular syndrome and accelerate vestibular compensation [82,83]. The second is the demonstration for the first time of reactive neurogenesis that ensures the production of new neurons in deafferented VN [84,85]. Pharmacological blockade of these newly generated neurons by in vivo infusion of an antimitotic agent significantly delays functional restoration, assigning this neurogenesis an adaptive status [86]. Again, pharmacological actions to stimulate the production of new neurons by proneurogenic agents such as BDNF have increased neurogenesis in deafferented vestibular nuclei and significantly accelerated vestibular compensation [80].

For years, we have been engaged in a research process aimed at identifying pharmacological pathways capable of stimulating neuronal excitability and reactive neurogenesis, two processes capable of attenuating the vestibular syndrome and optimizing the restoration of vestibular functions in animal models of vestibular pathology.

The intrinsic properties of T4 make this hormone a prime candidate. Indeed, T4 increases the expression of BDNF and KCC2, two important elements for the expression of neurogenesis and the maintenance of neuronal excitability [87], which are, moreover, two major actors of vestibular compensation [80,85,86]. Numerous studies have shown that thyroid hormones, including T4, regulate many aspects of neurogenesis, including proliferation, survival, migration, differentiation and maturation of neuronal and glial cells [2,88,89,90]. Furthermore, various studies indicate a modulatory role of thyroid hormones in neuronal excitability [91,92,93,94]. Finally, in the adult nervous system, T4 has been shown to have a neuroprotective effect and promote regeneration in experimental models of trauma [95].

Very few studies have investigated the role of thyroid hormones in vestibular compensation. One study shows that hypophysectomy delays vestibular compensation in developing tadpoles with unilateral labyrinthectomy [96]. However, thyroxine injections restore compensation kinetics in tadpoles without a pituitary gland. Another study shows that a continuous subcutaneous infusion of TRH for 14 days accelerates vestibular compensation in unilaterally labyrinthectomized adult monkeys [97]. Surprisingly, TRH receptors are present in the vestibular nuclei of adult rats [98,99], although their role remains unknown. The only two behavioral studies in the literature presented here have not elucidated the mechanisms of action of thyroid hormones underlying the observed effects. Our recent data shed new light on these mechanisms [81]. Indeed, in this work, we performed surgical unilateral vestibular neurectomy (UVN) on two groups of adult Long Evans rats (8 weeks) and for the first 3 days after UVN, we intraperitoneally injected one group with L-T4 (10 µg/kg) and the other group with a saline solution. As described by Yu and collaborators [100], we expected that the dose of L-T4 injected into UVN rats induced a transient hyperthyroid state a few hours after the first injection and then returned to a physiological state about a week later. We demonstrated that a short-term L-T4 treatment significantly reduced the vestibular syndrome evaluated with several vestibular tests (support surface area, weight distribution, qualitative vestibular syndrome scale, and open field) (for more details about the different tests, see: [81,101,102]). Two-way repeated measure ANOVA with post-hoc Bonferroni’s multiple comparison tests were used to determine statistical differences between the two groups. Briefly, compared to the control group, L-T4-treated rats showed an approximately 30% reduction in symptoms on the first day after the lesion following a qualitative assessment of the vestibular syndrome (behavioral symptoms of vestibular imbalance were scored for 10 components). We also demonstrated that L-T4 treatment promoted a faster recovery of postural and locomotor functions that had been impaired by UVN. The support surface area (or base support), which is increased after UVN and is a good estimate of postural stability and restoration of balance, is restored as early as the second post-lesion day in UVN rats treated with L-T4. Finally, the results of video-tracking assessments in the open-field arena revealed that the locomotion of UVN rats treated with L-T4 greatly improved. We demonstrated a significant improvement in walking speed and positive acceleration within the first days after the lesion in UVN rats treated with L-T4. We also highlighted the sites of thyroid hormone actions by demonstrating for the first time in adult rodents the presence of thyroid hormone receptors (TRα and TRβ), as well as the T4 to T3 converting enzyme (D2) within the vestibular nuclei. We will now discuss the different possible mechanisms of action of T4 underlying the obtained effects on vestibular compensation.

### 5.1. Search Strategy 

We conducted a literature search in PubMed up to 4 February 2023, using the following keywords: “Vestibular”, “Vestibular system”, “Vestibular disease”, “Vestibular disorders” “Vestibular compensation”, “Menière disease” and “Thyroid hormones”, “Thyroxine”, “TRH”, ‘Hypothyroidism”, “Hyperthyroidism”. Other possible articles were searched manually from the list of citations provided with each article. Articles published in the last 5 years were included as a priority. Both authors reviewed and selected abstracts that were relevant to each subsection of Chapter 5 of this study. 

### 5.2. Neuronal Excitability 

Restoration of a physiological level of excitability of the neural network in the deafferented vestibular environment is a crucial parameter for vestibular compensation [80,85,86]. Various studies indicate a modulatory role of thyroid hormones on neuronal excitability [91,92,93,94]. High doses of T3 or T4 have been shown to act through a direct action on GABA_A_ receptors by reducing GABAergic postsynaptic inhibitory currents in cultured hippocampal neurons [103,104]. T4 treatment produces a large increase in the resting discharge rate of neurons in the glomerular and subglomerular regions of olfactory bulbs [93]. Thyroid hormones have also been shown to facilitate neuronal excitability by increasing Na^+^ currents and the frequency of action potentials in the hippocampus and cortical neurons of rats after birth [91]. In the zebrafish model, the rapid increase in voltage-dependent sodium currents in neurons by T4 requires both αVβ3 and Nav1.6a membrane receptors [105,106]. Thyroid hormones have also been shown to increase the excitability of the peripheral nervous system (sciatic nerve) [92] and neuronal discharge rates in cats [107]. Finally, electrophysiological findings in an Alzheimer’s rat model demonstrated that thyroid hormones increased spontaneous neuronal activity in the dentate gyrus [94]. However, the exact molecular mechanisms of actions of thyroid hormones on neuronal activity remain to be elucidated. Taken together, these studies also support the non-genomic action of T4 and T3 thyroid hormones. Given the early effect of T4 treatment on vestibular compensation (as early as 1 day post UVN), it can be argued that in our UVN rodent model, thyroid hormones act through both genomic actions on TRα and TRβ targets and non-genomic actions on other receptors. 

### 5.3. Energy Metabolism 

Thyroid hormones play a key role in cellular metabolism and regulate numerous signaling pathways involved in carbohydrate, lipid and protein metabolism in several target tissues. Notably, hyperthyroidism induces a hypermetabolic state characterized by increased resting energy expenditure, reduced cholesterol, increased lipolysis and gluconeogenesis followed by weight loss, whereas hypothyroidism induces a hypo-metabolic state characterized by reduced energy expenditure, increased cholesterol, reduced lipolysis and gluconeogenesis followed by weight gain. Thyroid hormones also regulate respiration and mitochondrial biogenesis [9,108]. We will primarily focus on the effects of thyroid hormones on mitochondrial respiration and ATP production to interpret our behavioral results.

We demonstrated that short-term T4 treatment induces a bilateral increase in the number of cytochrome oxidase-positive neurons in the lateral vestibular nuclei 3 days after UVN [81]. Cytochrome oxidase is a Krebs cycle enzyme involved in ATP production. Thus, we can assume that stimulation of ATP production by thyroxin promotes the restoration of the neuronal excitability in ipsilesional vestibular nuclei, which is responsible for the acceleration of vestibular compensation. Also related to energy metabolism, we observed an absence of KCC2 downregulation induced after L-T4 treatment. This result is consistent with the hypothesis of Kaila et al. [109], who propose that “downregulation of KCC2 after neuronal injury may be part of a general adaptive cellular response that facilitates neuronal survival by reducing the energetic costs required to preserve low [Cl-]i”. Therefore, in our model, by providing thyroid hormones, known to increase energy metabolism, internalization of KCC2 to reduce energy costs after UVN is no longer required.

### 5.4. Adult Neurogenesis

Thyroid hormone signaling governs many aspects of neurogenesis, including neuronal and glial cell proliferation, survival, migration, differentiation, and maturation [2,88,89,90]. The precise mechanisms of actions of thyroid hormones on adult neurogenesis and the set of genes involved are still unknown, but the following three levels of actions can be described.

At the cellular level: In the subventricular zone (SVZ) of adult mice, hypothyroidism reduces the proliferation of neural stem cells (NSC) and neuronal progenitors by blocking re-entry into the cell cycle (Lemkine et al., 2005). In addition, the number of migrating neuroblasts to the olfactory bulb also decrease during hypothyroidism, showing that neurogenesis is globally impaired [11,12]. In contrast, hyperthyroidism in adult rats does not alter hippocampal progenitor proliferation, survival, or differentiation [110].

At the molecular level: It has been shown that TRα1 is not detected in NSC, whereas it is detected in neural progenitors and is highly expressed in neuroblasts. Furthermore, the overexpression of TRα1 in the subventricular zone niche promotes NSC commitment and differentiation to a neuroblast phenotype [12]. Thus, in the adult SVZ, T3 via the TRα1 receptor promotes the commitment of neural stem cells to a neuronal phenotype.

At the metabolic level: The effects of thyroid hormones on neurogenesis and the orientation of cell fate towards a neuronal phenotype could be induced by the impact of T4 and T3 on mitochondrial cellular respiration. It is indeed well established that thyroid hormones are crucial regulators of mitochondrial metabolism influencing ATP production, and thus cellular energy metabolism [111,112].

Adult neurogenesis is a multi-step process involving neural stem cell proliferation, survival of these cells, their differentiation into different cell types, maturation and functional integration into pre-existing neural networks [113]. Thyroid hormone signaling is involved in all these steps of neurogenesis [2,88,89,90]. Dutheil and colleagues demonstrated that by blocking or promoting cell proliferation, vestibular compensation was significantly delayed or accelerated [80,86,114]. Indeed, chronic infusion in adult cats immediately after UVN of the antimitotic cytosine-beta-D-arabino-furanoside (AraC) in the fourth ventricle completely blocks cell proliferation in deafferented vestibular nuclei. At the behavioral level, the recovery time of postural and locomotor functions was significantly delayed. Conversely, an infusion of BDNF under the same conditions from the first day after UVN increases the rates of cell proliferation, survival and differentiation. Under these conditions, animals significantly recover their balance and posture earlier [80].

Thyroid hormones promote the commitment of neural stem cells preferentially to a neural fate [12,89,90,115]. Thus, the increase in TRα+ cells observed as early as day 3 in deafferented vestibular nuclei and maintained until one month in thyroxin-treated UVN rats would have the following two effects on vestibular compensation [81]: (1)The increased probability of binding and uptake of endogenous and exogenous thyroid hormones in the deafferented vestibular nuclei, increasing the energy metabolism of neurons and glial cells.(2)The activation of quiescent vestibular stem cells [81,116] (which overexpress TRα), potentially promoting the formation of new neurons contributing to vestibular compensation and its maintenance over time.

However, in our paradigm, cell proliferation and survival were significantly increased and cell differentiation was preferentially directed towards a microglial rather than neuronal fate in L-T4 treated rats. This result corroborates with another study from our group showing that a sensorimotor rehabilitation protocol also promotes microglial differentiation in the UVN rodent model [117]. These data show that microglia represent a common target for LT4-based pharmacological treatment and sensorimotor reeducation. 

### 5.5. Potentiation of T4 to T3 Conversion by the Glial Response Produced by Vestibular Neurectomy 

The fact that the T4 to T3 converting enzyme is mainly expressed in astrocytes is an important point. Animal models of vestibular damage (neurectomy and surgical vestibular labyrinthectomy) show strong astrocytic responses in the deafferented vestibular nuclei [80,86,114,116,118,119,120,121]. T4-treated UVN rats also show strong astrocytic responses in the deafferented vestibular nuclei [81]. Therefore, the conversion of T4 to T3 could be exacerbated after UVN in the vestibular nuclei and could promote all mechanisms favorable to vestibular compensation (energy metabolism, excitability, neurogenesis etc.). Moreover, if a treatment of exogenous T4 is administered to the animal, we would observe a combination of endogenous T4 plus exogenous T4, whose conversion into active T3 would be further enhanced by the strong astrocytic reaction and explain the beneficial effects on functional restoration. Consistent with this hypothesis, we have recently demonstrated that in a model of reversible partial unilateral deafferentation induced by transtympanic administration of kainic acid (TTK), the same dose of L-T4 injected in rats as in a UVN model does not show a beneficial effect on vestibular compensation [122]. Contrary to the UVN model, the TTK model does not induce a strong glial response in the deafferented vestibular nuclei [121]. This lack of astrocytic reaction could explain the absence of a beneficial result on vestibular compensation in the TTK model. If extrapolated to the human clinic, to obtain beneficial effects of thyroid hormone treatment, it would be important to target vestibular pathologies associated with central inflammation (e.g., vestibular neurectomy after schwannoma resection).

### 5.6. Myelination

The action of T3 on myelination is the best characterized effect in the literature [115,123,124,125,126]. In general, thyroid hormones regulate oligodendrocyte differentiation and myelin production via transcriptional effects [127]. We have recently demonstrated in deafferented vestibular nuclei an increase in oligodendrocyte numbers 3 days after UVN. Furthermore, 1 month after injury, oligodendrogenesis is present in the deafferented vestibular nuclei [116,117]. The proliferation and recruitment of oligodendrocytes in the vestibular nuclei suggests a role in synaptic reorganizations, supporting vestibular compensation. One possibility is that remyelination of different axons within the vestibular network may promote vestibular compensation. These neural networks may have an intrinsic origin within the vestibular nuclei with the endogenous vestibular neurogenesis and myelination of locally newborn neurons. However, they can also have an extrinsic origin, such as dentritic spine growth and axonal sprouting from afferent systems in the vestibular nuclei. Indeed, structural changes, such as reactive synaptogenesis or axonal sprouting, also occur in the deafferented vestibular nuclei [128,129]. This hypothesis would be in favor of sensory substitutions [76]. The synaptic weight of visual, tactile and proprioceptive sensory inputs within the vestibular nuclei would be strengthened in order to replace the primary vestibular inputs suppressed by the deafferentation. The purpose of these sensory substitution processes is to restore a physiological level of excitability within the deafferented vestibular nuclei, but also to maintain vestibular compensation and the various adaptive strategies put in place to maintain posture, balance and gaze stabilization.

### 5.7. Angiogenesis

Cell migration and recruitment of different cell populations occur via the cerebrospinal fluid or the vascular network. Therefore, angiogenesis may be a contributing factor to the structural and functional reconfiguration of the deafferented vestibular network. Indeed, therapeutic angiogenesis has been used to improve brain plasticity by promoting the formation of new blood vessels and restoration of blood flow to the damaged area [130,131,132]. The angiogenic effects of thyroid hormones have been demonstrated in myocardial tissues [133]. In addition, an increased number of new blood vessels was demonstrated in the brain of hypothyroid rats after administration of a T4 analog [134]. Our observations (unpublished data) show a much more vascularized deafferented vestibular tissue after UVN in rats and cats. It would have been interesting to use specific markers of the vascular network to be able to stain and quantify the vascular arborization in the vestibular nuclei after UVN and T4 treatment.

### 5.8. Muscle Tone

Skeletal muscle is a primary target of thyroid hormone signaling. Indeed, the thyroid hormone transporters MCT8 and MCT10, as well as the thyroid hormone-converting enzymes D2 and D3, are expressed in skeletal muscle in humans and rodents. Thyroid hormones participate in (i) contractile function by increasing the rate of muscle relaxation-contraction, (ii) metabolism by increasing mitochondrial biogenesis, (iii) myogenesis and muscle regeneration [135,136,137]. The presence of thyroid hormone receptors in extraocular and skeletal muscles [45] could imply an action of thyroid hormones in gaze stabilization and in the recovery and maintenance of antigravity muscle tone (in the soleus muscle for example). UVN induces muscle tone asymmetry with ipsilesional hypotonia and contralesional hypertonia [138,139], altering postural and locomotor balance function. Although no study to date has demonstrated an increase in muscle tone after thyroid hormone injection, it can be assumed that by increasing contractile function and muscle metabolism, thyroid hormones may promote the rebalancing of muscle tone after vestibular deafferentation, thus accelerating the restoration of static and dynamic equilibration function. It would be interesting to perform an electromyogram of antigravity muscles in thyroxine-treated neurectomized rats. Similarly, it would be interesting to measure the effects of acute T4 treatment on ocular nystagmus kinetics and vestibulo-ocular reflex compensation.

### 5.9. Interaction between Histamine and the Thyroid Axis

It is now well established that histamine influences vestibular compensation [73,140,141,142]. Betahistine, an antivertigo drug that has been used for many years, leads to an increase in histamine synthesis and release as a histamine H1 agonist and by a specific blockade of the H3 autoreceptor [55,143,144,145,146,147,148]. Histamine, through its depolarizing action on H1 and H2 receptors carried by vestibular neurons, would restore electrophysiological homeostasis between intact and deafferented VN, thus facilitating functional restoration [55]. In regard to thyroid hormones, it is interesting to mention that histamines control TSH release [149], suggesting a possible role for histamine in thyroid hormone regulation (for review see [150]). In return, Upadhyaya and colleagues demonstrated in rats treated with L-T4 that histamine levels were increased in the hypothalamus, thalamus, and cortex, and that circulating T3 and T4 levels were positively correlated with histamine [151]. These results suggest that histamine and thyroid hormones interact with each other upon their respective release. Thus, can we assume in the context of a vestibular pathology that a treatment with thyroid hormones can increase the release of histamines, and thus promote vestibular compensation? Vestibular lesions themselves induce an increase in histamine synthesis in neurons in the tuberomammillary nuclei and release in the vestibular nuclei [55,144,145] and histamines actively participate in the promotion of vestibular compensation. A better understanding of the mechanisms underlying the interaction between histamine and thyroid hormones may provide new therapeutic alternatives for the treatment of vestibular pathologies.

### 5.10. Thyroid Hormones, Emotional Component and Vestibular Compensation

Stress hormones such as cortisol and adrenocorticotropic hormones have been reported to be repeatedly elevated in dizzy patients [152,153,154]. Stress is an important factor in vestibular compensation, as patients with chronic stress have a poor prognosis for functional recovery [51,52]. Yet, serotonin is a crucial neurotransmitter in the inhibitory control of stress [155,156], which is still very poorly considered in the understanding of vestibular compensation. The use of thyroid hormones as an effective adjunctive treatment for affective disorders has been studied over the last three decades and has been confirmed repeatedly [157]. The interaction of the serotonin system with thyroid hormones has been suggested as a potential underlying mechanism of action [158]. In the majority of studies, the effects of thyroid hormone administration in animals with experimentally induced hypothyroid states include an increase in cortical serotonin (5HT) concentrations and desensitization of autoinhibitory serotonin 5-HT1A receptors in the raphe region (the source of serotonin-producing neurons), resulting in increased serotonin release in the cortex and hippocampus. In addition, there is evidence that thyroid hormones may increase the sensitivity of cortical 5-HT2 serotonin receptors [159]. Therefore, it can be suggested that exogenous T4 in vestibulo-lesioned rats may enhance serotonergic neurotransmission via desensitization of 5-HT1A receptors in the raphe region, and increased sensitivity of 5-HT2 receptors. This action of T4 on serotonin would attenuate the stress induced by the vestibular syndrome, which would probably benefit the restoration of the balance function. 

A summary diagram (Figure 5) of the different possible pathways of action of thyroid hormones on vestibular compensation is presented below.

## 6. Conclusions

In summary, our previous work demonstrated that the short-term administration of L-T4 after unilateral vestibular loss greatly reduces vestibular syndrome and improves vestibular compensation. These results open new perspectives of L-T4 treatment for acute vestibular loss. According to our knowledge, there are no existing high-quality studies investigating the effect of an application of thyroid hormones in patients with acute unilateral vestibulopathy. However, there are arguments that make a translation of results from preclinical studies to proof-of-concept trials seem realistic. (1) Previous studies have demonstrated considerable similarities in the mechanisms of vestibular compensation across species [160]. In humans, static symptoms of acute unilateral vestibulopathy can be resolved within days or weeks, as in rodents. (2) Thyroxine is one of the most frequently prescribed drugs in the world, which is widely available, cheap and safe, if contraindications are considered. Its application in the context of acute unilateral vestibulopathy is likely to be restricted to the critical phase of vestibular compensation, i.e., to the first 1–2 weeks after symptom onset.

More broadly, the hormonal system and the vestibule seem to be intimately linked in regard to the presence of several receptors within the vestibular apparatus or the central vestibular nuclei. It seems then important to more extensively investigate the link between the endocrine system and the vestibule in order to better understand the vestibular physiopathology and to find new therapeutic leads.

## Figures and Tables

**Figure 1 ijms-24-09826-f001:**
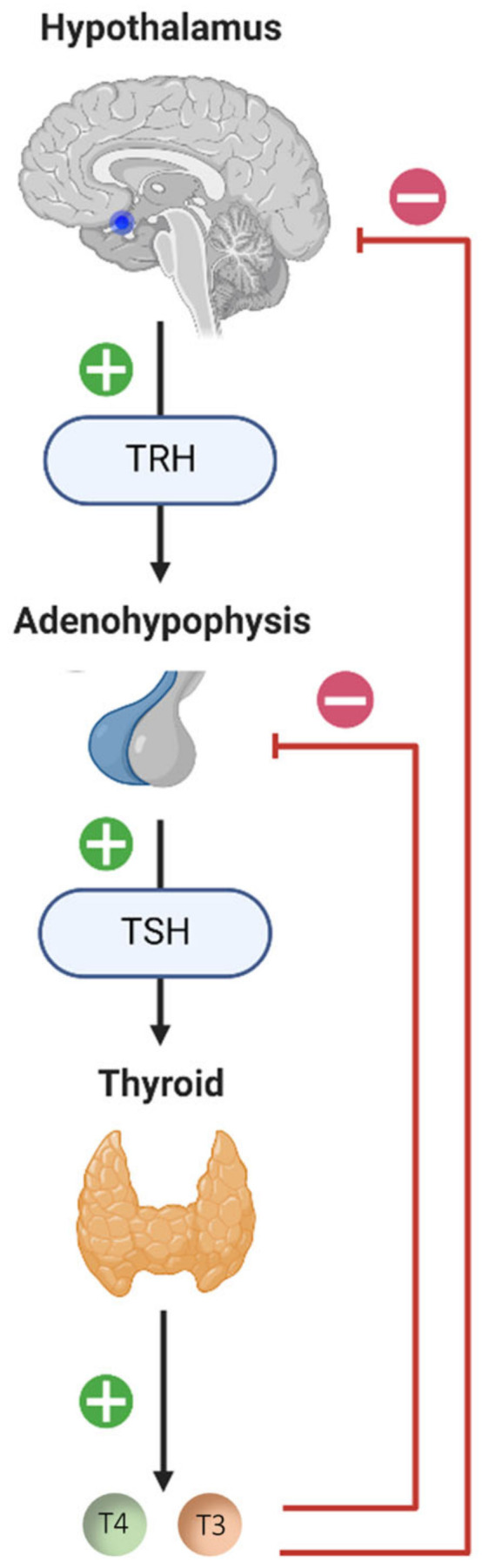
**Diagram of the hypothalamic–pituitary–thyroid (HPT) axis**. The neurons of the paraventricular nuclei of the hypothalamus synthesize and secrete TRH (Thyrotropin-Releasing Hormone), which stimulates the release of TSH (Thyroid-Stimulating Hormone) by the adenohypophysis. TSH will bind to its membrane receptor present in the follicular cells of the thyroid and cause the synthesis and release of thyroid hormones: T4 (Tetraiodothyronine, Thyroxine) and T3 (Trioodothyronine). High plasma concentrations of thyroid hormones (T4 and T3) exert negative feedback from the pituitary and hypothalamus, negatively influencing their own secretion. The blue dot in the sagittal section of the brain diagram represents the location of the hypothalamus. The + signifies stimulation and the − inhibition.

**Figure 2 ijms-24-09826-f002:**
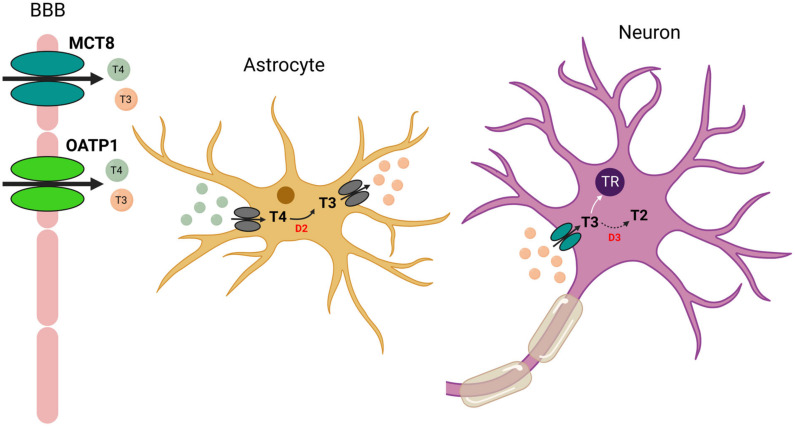
**Proposed model of astrocyte–neuron interaction and T3 transport in the brain.** T4 (and to a lesser extent T3) is transported across the blood–brain barrier (BBB) by OATP1 or MCT8. T4 is then transported into astrocytes by a still unknown transporter. In astrocytes, T4 is converted to T3 by the enzyme D2. Biologically active T3 is then transported out of astrocytes by another unknown transporter. Neuronal uptake of T3 is facilitated by MCT8, and then T3 exerts its genomic action by binding to its nuclear receptor (TR). Finally, T3 is inactivated by D3 (Iodothyronine Deiodase type 2) to T2 (diiodothyronine).

**Figure 3 ijms-24-09826-f003:**
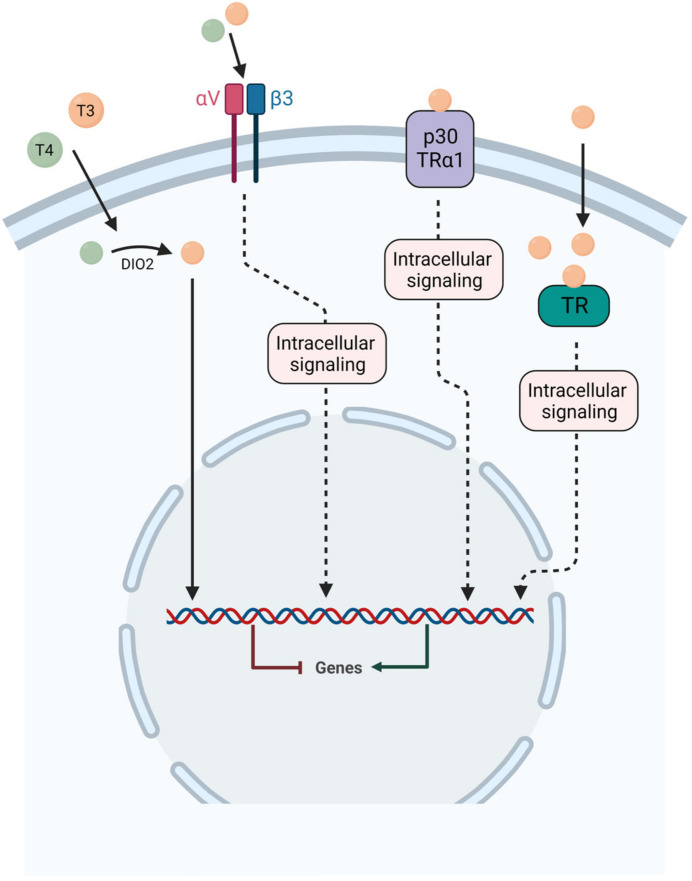
**Schematic representation of both genomic and non-genomic actions of thyroid hormones.** For genomic action, T4 and T3 are transported inside the cell and T4 is converted to T3 by the enzyme D2, then T3 acts through specific nuclear receptors TRα and TRβ to modulate gene expression. For non-genomic actions, T4 and T3 can act at the plasma membrane on the integrin αVβ3 receptor or with p30 TRα1. In the cytoplasm, T3 can act on both thyroid hormone receptors (TRα and TRβ).

**Figure 4 ijms-24-09826-f004:**
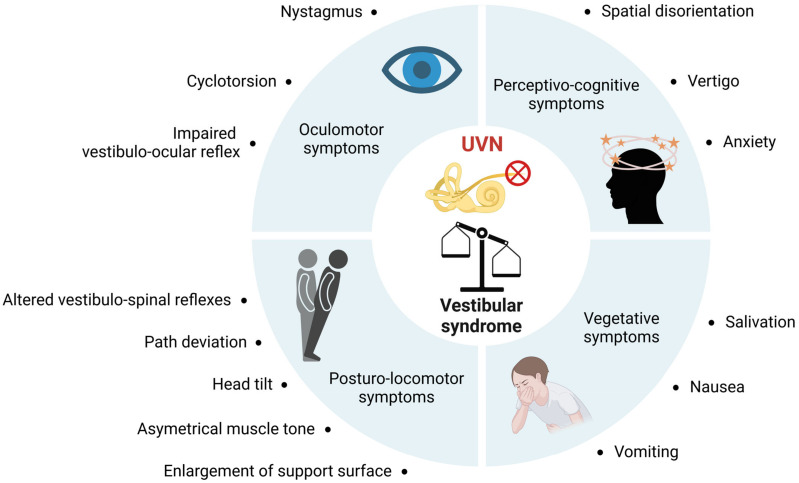
**Summary of functional consequences of unilateral vestibular neurectomy**. Unilateral vestibular nerve section induces a quadruple syndrome: oculomotor, posturo-locomotor, vegetative and perceptivo-cognitive.

**Figure 5 ijms-24-09826-f005:**
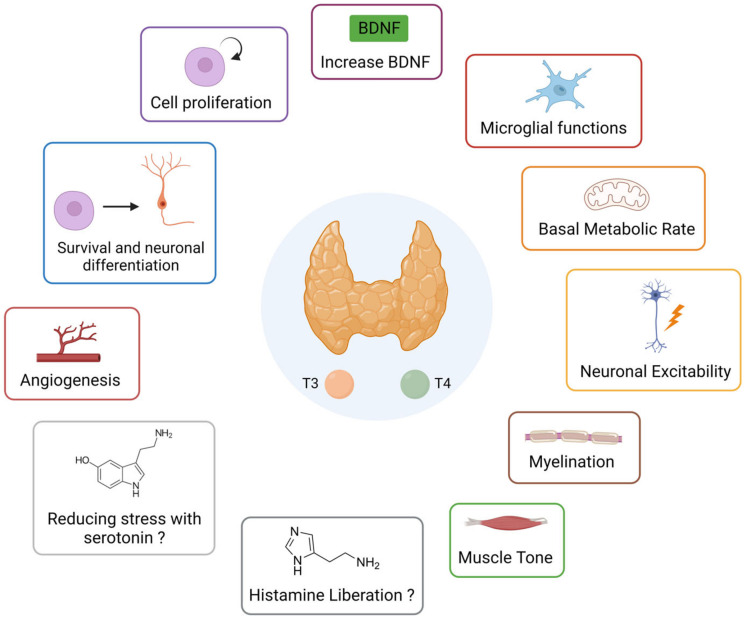
Summary diagram of the different possible actions of thyroid hormones that can promote vestibular compensation.

## Data Availability

Not applicable.

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
