# Peer review of "Thyroid Axis and Vestibular Physiopathology: From Animal Model to Pathology"

_ijms, 2023, doi:10.3390/ijms24129826_

Round 1

Reviewer 1 Report

In the article, the authors reviewed the effect of thyroid hormones on vestibular pathophysiology. The introduction part of the article is mostly based on the regulation of thyroid hormones and their mechanism of action in the central nervous system. Then, the relationship between inner ear development and balance systems and thyroid diseases was tried to be revealed. Here, rather than a general evaluation, Hashimoto's and Meniere's diseases are mentioned. I am in the opinion that the part of the article numbered as chapter 5 is more interesting. Here, it is aimed to shed light on future studies by examining the possible mechanisms of action of thyroid hormones on vestibular compensation.

The lack of methodology as a review stands out as the most important deficiency of the article. Information such as how the articles cited in the review were selected, according to which criteria the selection was made, which databases were used to search for articles will increase the value of the review.

I think that if the information up to the 5th chapter is shortened by focusing more on the vestibular system and the methodology is added for the 5th chapter, it will be an article that will contribute to the literature.

Author Response

First, the authors are grateful to the reviewer for giving us the opportunity to revise and improve the paper while offering his/her helpful and constructive criticism.

In agreement with your request, we have added in Chapter 5 a method sub-section concerning the articles cited in the review. You will find this paragraph in red between line 327 to 335 of the revised manuscript.

Reviewer 2 Report

Rastoldo and Tighilet conducted a comprehensive review of thyroid axis and vestibular physiopathology. The review is well-written but I have some concerns on the physiological role of thyroxin on the vestibulopathy and vestibular functions.

1.     The conclusion is heavily based on their recent publication in a rat model of acute peripheral vestibulopathy where they demonstrated that a large dose of thyroxine treatment (10 μg/kg for 3 days) improves vestibular function. However, the dosage used is significantly higher than that used for thyroid hormone replacement therapy to treat patient with congenital hypothyroidism (1.6 μg/kg), considering the rat still has normal thyroid hormone production after UVN (unilateral vestibulocochlear nerve) section.  Therefore, the observed thyroxin effects on vestibular functions may not be physiological effects.

2.     The dosage (10 μg/kg) would likely result in hyperthyroidism. Did the authors study the rat pituitary-thyroid axis after thyroxine treatment: serum T4 and TSH level?

3.     What is the clinical implications or translations for Meniere's disease?

Author Response

We would like to thank the reviewer for the time spent proofreading the article and for his/her constructive criticism. To respond to your comments:

  1. The dose used in our study was based on the dose used in the study of Shabani et al. 2016 (DOI: 10.1016/j.npep.2016.09.003) and Bavarsad et al. 2018 (DOI: 10.1080/01480545.2018.1481085). The low dose of L-T4 (10µg/kg) used in rats during 3 weeks in the study of Bavarsad and colleague demonstrated the best effect compared to the high dose (100µg/kg). Furthermore, it is interesting to note that the metabolism of thyroid hormones in rats is different from that in humans.. As demonstrated by the study of Capen (DOI: 10.1177/019262339702500109), the plasma T4 half-life in rats is considerably shorter (12-24 hr) than in humans (5-9 days).

As the reviewer noted, we do not know whether unilateral vestibular neurectomy per se alters thyroid hormone production and this is a point already discussed in chapter 3 of the present review. Whether the observed effect is due to a physiological dose of injected L-T4 or not, in our UVN model we still observed a significant effect on functional recovery and we discussed this effect extensively in this review.

  1. Unfortunately, we did not study the rat pituitary-thyroid axis after thyroxine treatment. However, as pointed by the reviewer and based on Bavarsard's study, we can assume that the dose of 10µg/kg in our model induces a transient hyperthyroidism, which will be regulated by the rats. Indeed, Yu et al., (DOI: 10.1016/j.yhbeh.2015.01.003) demonstrated that a single injection of L-T4 of 5, 15 of 20 µg/kg in adults rats significantly increased serum concentration of FT4 at 24h post-injection. However, seven days after a single injection of L-T4 (15µg/kg) serum FT4 and FT3 return to control levels.

  1. Very interesting question but somewhat difficult to answer. Indeed, autoimmune factors, trauma, viral infection, genetic predisposition, hormonal disorder, and metabolic factors could contribute to the development of Meniere's disease and each of these factors must be considered in order to propose an appropriate treatment for the pathology. As discussed briefly in the chapter 4 of the present review, several studies demonstrated that both hypo and hyperthyroidism were related to Meniere’s disease. However, the attenuation of Meniere's disease after thyroxine supplementation of hypothyroid patients is controversial. However, what if an euthyroid patient with Menière's disease is being treated with L-thyroxine? Based on our behavioral results on the rodent model it seems that for thyroxine to accelerate functional recovery, there must be a central inflammation in the vestibular nuclei. We have shown that the effects of T4 are effective in the UVN rodent model generating inflammation in the VN (Rastoldo et al., 2021) whereas they are minimal in a non-inflammatory vestibulopathy model (Bringuier et al.,2022). To come back to the factors that can lead to Menière's disease, we think that if the patient has Menière's disease with inflammation, then treatment with L-T4 could be favorable. This hypothesis still needs to be verified.

Reviewer 3 Report

The hypothalamic-pituitary-thyroid axis is one of the most important neuroendocrine pathways regulating metabolism and stress response, while proper vestibular functions are essential for our daily activities. Understanding of the interactions between those two systems is important for us to modulate one system by manipulating the other system for therapeutic purposes. This manuscript “Thyroid axis and vestibular physiopathology: from animal 2 model to pathology” by Rastoldo and Tighilet is an excellent review article summarizing recent findings in the relationship between thyroid hormones and the vestibular system with animal models and human patients. The manuscript was well written, the sections were organized in a way easy to follow, and the literatures cited were sufficient. On top of literature discussion, the authors speculated a few mechanisms for further investigation, which makes this article more interesting to read. The only issue I found is that there are so many abbreviations, and it would be helpful to have a table for all the abbreviations. Also, there are a few minor edits required: 1) what does the word “PVN” in line 184 stand for, paraventricular nucleus? 2) subventricular zone is usually abbreviated as SVZ not ZSV, in line 375; 3) NVU in lines 389 and 393 should be UVN.

Author Response

First, we would like to thank the reviewer for his careful evaluation of our manuscript and for his/her positive statement. We greatly appreciate that you found our review interesting and easy to read.

In accordance with the reviewer remark, we have added a table for all the abbreviations at the beginning of the revised manuscript.

To respond to your comment, Yes PVN stand for paraventricular nucleus and we added the abbreviation line 189.

We thank you for noting the typos that we have corrected in the new version of the revised manuscript. 

Reviewer 4 Report

Dear Authors,

I read your manuscript with great interest. The topic is exciting, so thank you so much for your great scientific review work and discussion on the relationship between thyroid and vestibular function. I think your work needs to be published, but it needs major revision. 

1. The abstract should be clarified and divided into sections, so that readers can quickly understand the content of the study: a. Aim or object of the study; B. Materials and methods; c.Results, d. Discussion; e. Conclusion.

2. Your introduction is complete, but a bit long. It might be condensed, but that's okay.

3. You must clearly describe the methods of your study. You describe it quickly from line 298 to line 309, but you should better specify your sample (young/old rats? Genetically identical? Unilateral deafferentation methods – chemical/surgical), national/international ethics committee clearance (whether it is expected or not and why) …

4. You must describe in detail your results, the methodological test you applied, the type of statistical data analysis you used.

5. Discussion is good, but you could condense it.

6. The conclusion must be focused on the results of your study. Afret that, you could present your personal patho-physiological hypothesis and personal conclusion of your study.

 Thanks to these corrections, I am sure you will add great scientific value to the excellent topic of your manuscript.

 Looking forward to reading your work review.

Best regards.

Author Response

First, the authors are grateful to the reviewer for giving us the opportunity to revise and improve the paper while offering his/her helpful and constructive criticism. We greatly appreciate that you read our review with great interest and that you liked the topic.

  1. Thanks to the reviewer's comment, we have rewritten the abstract in the revised manuscript.
  2. We understand the reviewer's point, and we prefer to leave the introduction as it is in order to have a comprehensive view of the hypothalamic-pituitary-thyroid axis and the pathways of action of thyroid hormones at the level of the central nervous system.

3& 4. Thanks to the reviewer’s comment we have more fully described the methods and results of our study. You can find the modifications in red between line 305 to 321 of the revised manuscript.

  1. We understand the reviewer's comment, indeed our discussion is quite extensive. However, this choice is voluntary. Contrary to a more concise discussion that we presented in our article (DOI: 10.3390/cells11040684), we wanted to address in this review other themes and hypotheses with a broader scope than what we were able to address in our scientific article.
  2. Thanks to the reviewer’s comment we modified the conclusion of the review and we focus first on the results of our previous study. The modifications are in red between line 582 to 585.

Round 2

Reviewer 1 Report

Thank you for the revision. The points that I focused on have been answered and added to the text. I believe that the article as it stands is sufficient for acceptance.

Author Response

We would like to thank the reviewer for his/her careful evaluation of our manuscript and for his positive statement.

Reviewer 2 Report

Authors now confirmed that the dosage used in the study of thyroid axis and vestibular physiopathology will cause hyperthyroidism in the rat model. The observed effects of T4 on the vestibulopathy and vestibular functions are actually derived from non-physiological dose of thyroxin, which raises the question of usefulness of their data. Since the subject of the review is about thyroid axis and its impact on vestibular physiology, it is misleading to use data generated from toxic dose of T4 to relate the impact of physiological T4 on vestibular function.

 Authors need to mention and discuss about this in the review.

Author Response

To induce hyperthyroidism Kim and Lee (https://doi.org/10.1155/2019/3239649) used daily subcutaneously injection with L-T4 at a dose of 0.3 mg/kg for 2 weeks. In Hwang et al., study (DOI: 10.1186/s12906-017-2036-1) hyperthyroidism was induced by intraperitoneal injection of LT4 (0.5 mg/kg during 4 weeks). Ahmed et al., (DOI : https://doi.org/10.15419/bmrat.v5i12.506) induced hyperthyroidism by intraperitoneal injection of LT4 at 100mg/kg during 2 weeks. In all studies investigating hyperthyroidism, nearly the same high dose of LT4 is injected into rats for 2 to 4 weeks: this is the standard method. As a reminder, in our study, we injected rats with LT4 (10µg/kg) daily for only 3 days, which is well below the dose used in hyperthyroidism studies. Now, how do you define hyperthyroidism? It is characterized by abnormally high levels of thyroid hormone, like free triiodothyronine (FT3) and free thyroxine (FT4). As explained by Yu and collaborators (https://doi.org/10.1016/j.yhbeh.2015.01.003) following a single injection of LT4 (15µg/kg) serum FT4 and FT3 were significantly increased 2 h post-injection but returned to baseline levels 7 days later. This hyperthyroid states is transient and cannot be attributed to hyperthyroidism as described in the other articles cited above. It is the same logic when we ingest sugar, we certainly induce a hyperglycemic state but the organism regulates it normally and returns to control values without having chronic hyperglycemia with all the related negative consequences.

Among all the symptoms caused by hyperthyroidism, hyperthyroid states in rats is accompanied by a decrease in body weight. However, in our UVN model, rats injected with L-T4 slightly decreased their body weight (less than 10%) only the second days after the surgery and then regained weight normally (data not shown). With almost no weight loss and no mortality observed in rats treated with L-T4 at 10µg/kg, we are nowhere near a toxic dose. We can mention in our manuscript that our L-T4 dose induced probably a transient hyperthyroid state (on the basis that vestibular lesion itself does not decrease FT4 but we do not know this) but we will not go any further in the discussion on this subject. You can find the modification on the revised manuscript line 308 to 311.

Reviewer 4 Report

2nd review

Dear Authors, Thank you so much for the revised version of your interesting manuscript.
You can find my point-by-point answer below.
I want to clarify that I really appreciate your work and I will do everything
to facilitate its publication. My suggestions are meant to add scientific value
to your manuscript, with scientific editing as appropriate as possible.
Point 1. I expected to see the abstract divided into sections, as I suggested,
to improve quick understanding of the manuscript. However,
the "descriptive" version you propose is clear and provides Readers
with any information necessary to understand purpose, results and
conclusions of your study, so I can accept it.
Point 2. I fully understand your intention and, given the importance and
scarcity of literature on the topic, I can accept your long introduction.
Points 3-4. Thanks for the revisited part, which makes it easier to understand
how you conducted the study. However, there are still some unclear points:
A. Have you followed/obtained any type of clearance/authorization from the
national/international Ethics Committee on animal experimentation or is it
not required? (Sorry, I have little experience with animal experiments and
regulatory data, so thanks for the clarification).
B. Now I understand the structure of your manuscript: it is descriptive and
focused on the relationship between thyroid and vestibular function,
collaterally supported by the experimental results you obtained on rats, but
not focused on this (otherwise the structure of the article should be centered
on your experimental data on rats and sections divided into ABSTRACT,
INTRODUCTION, MATERIALS AND METHODS, RESULTS, DISCUSSION,
CONCLUSION, REFERENCES). For this reason, I can accept the main
structure of your article in chapters, within which you describe your experiment
on two groups of vestibular deafferent rats, one treated with thyroid hormones
and the other with placebo.
C. Could you clarify what is indicated in lines 310-311: "different vestibular
tests (footprint, weight distribution, qualitative scale of the vestibular syndrome and open field)"? How are these tests conducted? Can you describe the evaluation and execution procedures? Did you obtain statistically comparable numerical data of each of the indicated parameters? Are there any references you can provide for readers who (like me) are new to this type of assessment? As an otoneurologist (and as many Readers will be), I am very curious and interested in fully understanding the experiments and thank you for your valuable contribute!!
Point 5. I fully understand your intention and, given the importance and
scarcity of literature on the topic, I can accept your long introduction.
Point 6. Thank you for the changes. We now well understand the importance
of your study which opens up new important research scenarios for the
treatment of an acute vestibular loss, frequent in the general population and
continuously increasing at any age.
I sincerely thank the Authors for the new effort required, but I believe it’s worth
it, because I am sure that your work could have an important resonance
in future scientific research on this topic and perhaps also in the clinical
treatment of an acute vestibular deficit.
I look forward to receiving your point-by-point answer.
Best Regards

Author Response

Dear reviewer, we believe that your pertinent observations were very constructive and helped us to improve the manuscript. We will do our best to answer the unclear points you raise and hope that our answers will satisfy you.

Point A: To perform experiments on animals and publish our study on thyroxine and vestibular compensation (DOI: 10.3390/cells11040684) we indeed obtained an evaluation and authorization from the national Ethics Committee on animal experimentation which is mandatory for studies involving animals. This part is present in the methods section of our study that you can find here: “All experiments were performed in accordance with the European Union 2010/63/EU Directive and under the veterinary supervision and control of the National Ethical Committee (French Agriculture Ministry Authorization: B13-055-25). The present study was specifically approved by Neurosciences Ethic Committee N°71 of the French National Committee of animal experimentation.”

In the present review, we do not think it is necessary to mention the ethical part of our previous study because interested readers can easily find this part in the study in question. However, if you think it is important to specify this part for the convenience of otoneurologists we would be happy to add it to the manuscript.

Point B: We are glad that you understand the structure of our manuscript with the modifications we made in the previous revision.

Point C: In our previous study, we used a variety a vestibular tests to evaluate vestibular compensation and the effect of a short-treatment of thyroxine. To best answer your question, we will review each test used.

The support surface area is a sensitive parameter used to assess static postural instability in cats or rats model of unilateral vestibular loss. It is similar to static posturography that quantifies an individual’s balance behavior in upright stance. However, we do not measure the center of pressure and body sway but the support’s surface delimitating the four paws of the animal. In some human study, the support surface area is reported as “the base support”. To quantify the support surface, rats were placed in a device with a graduated transparent floor that allowed them to be filmed from underneath. A scale drawn on the bottom served to take measurements of the location of the four paws. We only measure the area of support surface to the light, we do not have, as in human studies, different conditions with or without light (there is no condition “eyes closed”). To avoid measures when the animals were moving, we measured the support surface area when the animal landed. For this purpose, we picked up the animal by the tail and lifted it vertically to a height of about 50 cm (lift duration 2 s; position holding at upper position: 1 s) and dropped it to a height of about 10 cm. When the animal touched the ground, we took a capture of the location of the four paws. Twenty repeated measurements were taken for each rat tested at each time point (pre-lesion, D1, 2, 3, 7, 10, 14, 21, and 30 post-lesion), and an average was calculated for each experimental session. As with patients with an unstable gait, a wider support surface provides more stability for the rats. As you can find on our previous study, unilateral vestibular lesion results in an increase in the support surface delimited by the animal’s four paws. We observed that the UVN-NaCl group had significantly increased their support surface area the first 3 days following vestibular lesion (D1 to D3: p < 0.001) and reached pre-operative control values from D7 until D30. In contrast, the UVN-T4 group significantly increased their support surface area only at D1 after vestibular lesion (p < 0.001) restoring to pre-operative control values from D2.

To quantify the postural syndrome following UVN we used a device (DWB1®, Bioseb, Vitrolles, France) measuring the weight distribution at all contact points of the animal’s body with the ground. For each acquisition session, the rat was placed in the device for 5 min and could move freely. The device consists of a Plexiglas cage (25 × 25 cm) and the floor of this cage is fully covered with a plate with 2000 force sensors that detect vertical pressure at a sample rate of 30 Hertz. Based on previous results (DOI: 10.1371/journal.pone.0187472), we chose to analyze only the weight distributed on the lateral axis, which allows for the determination of how the animal distributes its weight between the left and right paws in order to maintain balance. As shown in previous reports (DOI: 10.1371/journal.pone.0187472; 10.3389/fneur.2020.00470) before UVN, rats distributed their weight symmetrically between the right and the left sides (about 50% of his weight in the right front and back paws and the other 50% in the left front and back paws). After UVN, UVN-NaCl rats increased the weight applied on their ipsilesional paws, resulting in a postural asymmetry that reached 8% at D7 and increased to 11% at D30 (D7: p < 0.05; D30: p < 0.001). In the UVN-T4 group, the postural asymmetry induced by UVN was not present.

We also performed a qualitative assessment of the vestibular syndrome. Behavioral symptoms of vestibular imbalance were scored for 10 components after UVN: the tail-hang test, rearing, grooming, displacement, head-tilt, barrel rolling, retropulsion, circling, and bobbing. Each component of the qualitative assessment have different score, and the higher the score, the more vestibular deficits the animal will have. You can find a video of circling behavior and bobbing behavior in this article: doi: 10.3389/fneur.2020.00505. In details:

-Tail hanging behavior: Animals were picked up from the ground at the base of the tail and body rotation was scored from 0 point (no rotation) to 3 points (several rotations of 360°)

-Landing reflex: After animals were picked up from the ground at the base of the tail, we scored the first 3 landings from 0 (presence of a landing reflex on the 3 landings) to 3 points (absence of landing reflex on the 3 landings). When lifted by the tail, control rats exhibit a landing reflex, consisting of forelimb extension, that allows them to land successfully (i.e., they land on all four legs). Rats with impaired vestibular function do not exhibit a forelimb extension, they spin or bend ventrally, sometimes “crawling” up toward their tails, causing them to miss their landings.

-Rearing: the ability of the rat to rear was scored from 0 point (rearing is observed) to 1 point (rearing is absent)

-Grooming: the ability of the rat to groom correctly were scored as follows: 0 point (correct grooming of full body) 1 point (grooming of the face, belly, and flanks but not the base of the tail), 2 points (grooming of the face and belly), 3 points (grooming of the face), 4 points (inability of the animal to groom itself)

-Displacement: quality of the displacement of the rat was scored from 0 (displacement of the rat with no visible deficit) to 3 points (several deficits in the displacement of the rat)

-Head tilt was scored by estimating the angle between the jaw plane and the horizontal with 0 points (absence of a head-tilt) to 3 points (for a 90° angle)

-Barrel rolling was scored as follows: 0 points (absence of barrel rolling), 1 point (barrel rolling evoked by an acceleration in the vertical axis of the rat in our hand), 2 points (spontaneous barrel rolling)

-Retropulsion characterizes backwards movements and was scored from 0 (absence of retropulsion) to 1 point (presence of retropulsion)

-Circling was scored from 0 point (absence of circling behavior) to 1 point (presence of circling behavior). Circling behavior refers to a repetitive walk of the animal in a circle. In the case of unilateral vestibular lesions, the animal will always turn in the direction of the lesion.

-Bobbing is related to rapid head tilts to the side and was scored from 0 point (absence of bobbing) to 1 point (presence of bobbing)

With this qualitative assessment we observed in both the UVN-NaCl and UVN-T4 groups, a peak of vestibular score on day 1 (UVN-NaCl: 17 ± 0.53; UVN-T4: 12 ± 0.53) and vestibular deficits never completely disappeared at 30 days after the lesion. However, from D1 until D7, UVN-T4 treated animals had a lower vestibular score than UVN-NaCl animals (p < 0.0001).

Finally, to quantify the locomotor syndrome following UVN, we used an automated video tracking software (EthoVision™ XT 14, Noldus, Wageningen, The Netherlands) and referenced it to the Open-field test. Animals were individually placed in a square open box (80 × 80 × 40 cm, called “square open-field” in animal studies, hence the name of the test) and were allowed to move freely for 10 min. Their behavior was recorded for 10 min using a digital camera and analyzed with EthoVision™ XT 14 software. The videos of circling and bobbing behavior were recorded during a session of an open-field test if you saw them (in this paper doi: 10.3389/fneur.2020.00505). Based on previous results [doi: 10.3389/fneur.2020.00505], the mean distance travelled, meander, mean locomotor velocity, mean number of high accelerations (>50 cm/s2), and mean percentage of time spent immobile were quantified (for details, see [doi: 10.3389/fneur.2020.00505]). Briefly, in the open field test, UVN-T4 rats significantly reduced their total distance travelled only on D1 (p < 0.05). Similarly, the immobility time of UVN-NaCl rats increased the first 3 days after the lesion before returning to pre-operative control values whereas in the UVN-T4 group it increased only on D1 (p < 0.001) and returned to control values by D2. UVN-T4 rats did not present a reduction of velocity the first 3 days after UVN compared to UVN-NaCl rats. In the same way, the number of high accelerations of the UVN-NaCl group was almost absent the first 3 days (D1 to D3: p < 0.001). In contrast, the UVN-T4 group was still able to achieve high accelerations during the first 3 days post-UVN. Finally, vestibular disorders induce imbalance and stumbling, which leads to difficulty in walking straight. The ‘meander’ parameter from Ethovision XT14 allowed us to quantify the difficulty of walking. While animals injected with a saline solution had a drastically unstable gait on D1 post-UVN, animals injected with L-T4 did not exhibit significant gait instability.

All the explanations that have been provided are present in the method section of our previous article (doi:10.3390/cells11040684). We have tried to answer with as much detail as possible in this revision, however due to the length of the text we will not be able to import all of these explanations into this manuscript. That said, in order to take into account your comment and to help the reading of otoneurologists we have made a condensed version of these explanations that you will find in the manuscript between line 318 and 327. As you suggested, we also provided 2 different studies (in addition to our main study) for details of the different vestibular tests we used in our study (line 314).

Once again, we thank you for your valuable comments and the opportunity to improve our manuscript. It is important to us that this manuscript be read and understood by otoneurologists and we are pleased that you were able to provide your feedback. Do not hesitate if you have other comments or modifications to make to the manuscript!

Round 3

Reviewer 2 Report

The observed effects of thyroid hormone on the vestibulopathy and vestibular functions are due to large doses of T4 treatment. Authors stated that a short-term L-T4 treatment significantly reduced the vestibular syndrome such as improvement of locomotion, walking speed and positive acceleration within the first days of T4 treatment. How long do the observed effects last after treatment stopped? If the therapeutic effects are transient with large dose T4 treatment, then it is not practical for clinical translation. Long-term or repeated treatment with large dose of T4 would result in serious side effects.

Author Response

In order to answer your questions as fully as possible, we have asked for the opinion of neurologist Andreas Zwergal, with whom we published our study in 2022 on L-T4 and vestibular compensation in rats. Andreas Zwergal is the Director of the German Center for Vertigo and Balance Disorders (DSGZ) and an associate professor for Neurootology at the Ludwig-Maximilians-University (LMU) Munich. We hope you find these answers helpful:

"Following the concept of a critical phase of lesion-induced neuroplasticity, a transient application of L-T4 may be sufficient to both reduce the symptoms of acute unilateral vestibulopathy and augment vestibular compensation on the long-term. Therefore, application of L-T4 in patients most probably can be restricted to 14 days after symptom onset at the longest. Of note, high-dose L-T4 application protocols are well-established in the clinical setting of thyroid suppression scintigraphy since decades and were shown to be save without serious side effects (Ramos et al. 2000, doi: 10.1046/j.1365-2265.2000.00898.x). Commonly, a dose of 2µg/kg/body weight or 150-200µg/day is used for 10-14 days to suppress the normal thyroid function in humans. For comparison of doses across species, a dose of 2,2µg/kg in humans is equivalent to 20µg/kg in rats (Jahnke et al. 2004, DOI: 10.1289/ehp.6637). In the rat study by Rastoldo et al. 2022, L-T4 10µg/kg was applied, which would correspond to 1,1µg/kg in humans and would be quite in the range of the reported doses from thyroid suppression test protocols. For a clinical use, potential contraindications such as manifest hyperthyroidism, severe cardiac arrhythmias, cadiac failure or pregnancy needed to be excluded. In abscence of these contraindications, the temporary use of higher doses of L-T4 is safe and clinically feasible."

Reviewer 4 Report

Dear Authors,

thanks for your kind and exhaustive reply. It was exciting for me to read about it, especially concerning methods of balance assessment in animal models after UVN.

In particular, I appreciated your point-by-point answers, and references to your previous studies, denoting a great research experience in that field.

Thank you again for your interesting manuscript.

Best Regards

Author Response

Thank you for your kind review. We are pleased you enjoyed the article and our answers to your questions. We would welcome the opportunity to exchange ideas with you again about our research. 

Best regards,